# Chronic Myelogenous Leukemia with Double Philadelphia Chromosome and Coexpression of p210 and p190 Fusion Transcripts

**DOI:** 10.3390/genes13040580

**Published:** 2022-03-25

**Authors:** Samara Silveira da Cruz, Aline Damasceno Seabra, Lais Helena Rescinho Macambira, Débora Monteiro Carneiro, Patrícia Ferreira Nunes, Thais Brilhante Pontes, Fernando Augusto Rodrigues Mello-Junior, Lucyana Barbosa Cardoso Leão, Fernanda de Nazaré Cardoso dos Santos Cordeiro, Thiago Xavier Carneiro, Caroline Aquino Moreira-Nunes, Rommel Mario Rodríguez Burbano

**Affiliations:** 1Laboratory of Molecular Biology, Ophir Loyola Hospital, Belém 66063-240, Brazil; sami-silveira@hotmail.com (S.S.d.C.); line.seabra@gmail.com (A.D.S.); laismacambira@yahoo.com.br (L.H.R.M.); deboramonteirocarneiro@gmail.com (D.M.C.); dsnpatriciaferreira@gmail.com (P.F.N.); thaisbrilhante85@gmail.com (T.B.P.); fernando.mellojr@hotmail.com (F.A.R.M.-J.); rommel@ufpa.br (R.M.R.B.); 2Human Cytogenetics Laboratory, Biological Science Institute, Federal University of Pará, Belém 66075-110, Brazil; 3Department of Hematology, Ophir Loyola Hospital, Belém 66063-240, Brazil; lucyana_cardoso@yahoo.com.br (L.B.C.L.); fernandancscordeiro@hotmail.com (F.d.N.C.d.S.C.); thiagoxavc@gmail.com (T.X.C.); 4Pharmacogenetics Laboratory, Department of Medicine, Drug Research and Development Center (NPDM), Federal University of Ceará, Fortaleza 60430-275, Brazil; 5Northeast Biotechnology Network (RENORBIO), Itaperi Campus, Ceará State University, Fortaleza 60740-903, Brazil

**Keywords:** chronic myeloid leukemia, blast crisis, double Philadelphia chromosome, p210 and p190 coexpression

## Abstract

The Philadelphia (Ph+) chromosome, t(9;22)(q34;q11.2), originates from a chimeric gene called *BCR-ABL* and is present in more than 90% of CML patients. Most patients with CML express the protein p210 *BCR-ABL* and, with a frequency lower than 5%, express rare isoforms, the main one being p190. In the transition from the chronic phase to the blast phase (BP), additional chromosomal abnormalities, such as the presence of the double Ph+ chromosome, are revealed. Of the 1132 patients analyzed via molecular biology in this study, two patients (0.17%) showed the co-expression of the p210 and p190 isoforms for the *BCR-ABL* transcript, with the concomitant presence of a double Ph+ chromosome, which was observed via conventional cytogenetics and confirmed by fluorescent in situ hybridization. The *BCR-ABL*/*ABL*% p210 and p190 ratio increased in these two patients from diagnosis to progression to blast crisis. To our knowledge, this is the first report in the literature of patients who co-expressed the two main *BCR-ABL* transcript isoforms and concomitantly presented Ph+ chromosome duplication. The evolution from the chronic phase to BP often occurs within 5 to 7 years, and, in this study, the evolution to BP was earlier, since disease-free survival was on average 4.5 months and overall survival was on average 9.5 months. The presence of the p190 transcript and the double Ph+ chromosome in CML may be related to the vertiginous progression of the disease.

## 1. Introduction

One of the main characteristics of chronic myeloid leukemia (CML) is the occurrence of a reciprocal translocation called the Philadelphia chromosome (Ph+), which is formed by the fusion of the 3′ sequences of the *ABL1* gene (Abelson) mapped at 9q34 with the 5′ portion from the sequences of the *BCR* gene (breakpoint cluster region) located at 22q11, giving rise to a chimeric gene called *BCR-ABL*, t(9;22)(q34;q11.2), which is used in clinical CML as a diagnostic marker, and also for monitoring and prognosis [1,2]. The BCR-ABL oncoprotein exhibits deregulated tyrosine kinase activity, which promotes cell proliferation, decreases the adhesion of leukemic cells to bone marrow stroma, and protects leukemic cells from apoptosis [3,4,5,6].

CML has a worldwide incidence of 1 to 2 cases per 100,000 adults per year, corresponding to 25% of all leukemias. The mean age at diagnosis is 56 to 60 years and it is more frequent in males [7,8]. Normal course CML consists of three stages: the chronic phase, accelerated phase, and blast crisis. They begin in the chronic phase, which represents approximately 90% of cases at the time of diagnosis. As the pathology progresses, there may be an evolution to an accelerated phase and blast crisis, where there is a proliferation of immature cells, which accumulate in the bone marrow, leading to the release of blasts into the bloodstream [9].

The majority of CML patients presents p210 protein, which occurs in a region where there is a 5.8 kb break in the *BCR* gene, known as major-bcr (M-bcr), between exons 12 and 16 (known as b1-b5), with most breaks associated with M-bcr occurring between exon 13 and 14 (known as b2-b3). At the molecular level, there are different genomic breakpoints, which generate rare isoforms that are frequent in less than 5% of patients. Among these are the p190 (e1a2) isoforms in a region known as minor-bcr (m-bcr). The co-expression of p210 and p190 is rare in CML [10,11,12,13,14]. The p190 protein occurs in the majority of Ph+ positive patients with acute lymphoblastic leukemia [15].

The double Ph+ chromosome is one of the most frequent chromosomal aberrations in addition to the presence of the classic Ph+ chromosome (single) and is associated with an overexpression of the *BCR-ABL* protein. Other additional chromosomal alterations are also common, such as monosomy of chromosome 7, trisomy of chromosome 8, trisomy of chromosome 21, isochromosome 17q, the loss of the X chromosome in women, and the loss of the Y chromosome in men, and molecular alterations, such as mutations in *TP53, RUNX1*, and *IKZF1* genes and in the tyrosine kinase domain of the *BCR-ABL* oncoprotein, in addition to epigenetic alterations, may dictate the stratification of the prognosis and new treatment protocols for patients who progress to blast crisis [16,17,18].

The aim of this work is to describe clinical and laboratory alterations that may explain the adverse prognosis of two patients with CML who worsened through blast crisis, as revealed by the presence of a double Ph+ chromosome and with the concomitant coexpression of the p210 and p190 isoforms.

## 2. Patients and Methods

This study was approved by the Ethics Committee of Ophir Loyola Hospital, Belém-Pará (CAEE 9611320.0.0000.5550). The two patients, one male and one female, one 44 and one 45 years old, with no family history of onco-hematological diseases, were admitted to the Ophir Loyola Hospital, a cancer reference hospital in northern Brazil, between May 2017 and October 2020, where they were diagnosed with CML and started therapy with imatinib mesylate (400 mg/day).

### 2.1. Conventional Culture and Cytogenetics

Bone marrow samples were seeded in tubes containing Marrowmax™ Bone Marrow Medium (Invitrogen, Waltham, MA, USA). Following this, 24 h cultures were performed. Here, 0.1 mL of colchicine (0.0016%) was added to the cultures two hours before each collection. After this period, the cells were centrifuged (1000 rpm for 10 min), the supernatant was removed, and a hypotonic treatment with KCl (0.075 M) was started at 37 °C for 20 min. The cells were then centrifuged and fixed three times with methanol/acetic acid (3:1). Cell suspensions were placed on histological slides. The GTG banding technique used was the one described by Scheres et al. [19]. The identification and classification of chromosomes followed the recommendations of the International System for Human Cytogenetics Nomenclature [20]. Twenty metaphases from each of the two patients were analyzed.

### 2.2. Fluorescent In Situ Hybridization

The fluorescent in situ hybridization (FISH) method was performed on slides with cells fixed in methanol/acetic acid from all patients. A directly labeled dual-color Vysis LSI *BCR-ABL* Dual Color Fusion Translocation Probe (Chicago, Illinois, USA) was used. The slides were washed in 3x saline sodium citrate solution (SSC) and dehydrated in 70%, 80%, and 90% ethanol. The samples were then denatured with 75% formamide/2x SSC (pH 7.0) at 70 °C for 2 min and transferred to an iced ethanol (−20 °C) series at 70, 80, and 90%. The probe was denatured at 96 °C for 7 min. Then, 10 μL was applied to the slide under a glass coverslip. In situ hybridization occurred at 37 °C in a moist chamber overnight. Post-hybridization washing was performed, and the nuclei were counterstained with DAPI/antifade. The hybridization was visualized using a fluorescence microscope Olympus BX41 (Shinjuku, Tokyo, Japan) with a triple filter DAPI/FITC/TRICT and a system for capturing images and image analysis Applied Spectral Imaging^®^ (Carlsbad, California, USA). We appraised 200 nuclei and/or metaphases per slide and the signs checked were in agreement with the criterion of Hopman et al. [21].

### 2.3. Real-Time Quantitative Reverse Transcription of Polymerase Chain Reaction (qRT-PCR)

Molecular analysis was performed for complementary DNA synthesis and the determination of BCR-ABL transcripts for p190 and p210 isoforms. Commercial qRT-PCR One Step/TaqMan kits (Mobius Life Science, Pinhais, Brazil) were used, using the principle of oligonucleotide hydrolysis stained with two fluorophores. First, 20 mL of peripheral blood collected in a tube containing ethylenediamine tetra acetic acid (EDTA) was treated with hemolysis buffer (NH_4_Cl + NH_4_HCO_3_). RNA was stabilized in TRI Reagent Solution (Thermo Fisher Scientific, Waltham, MA, USA) and isolated with 100% isopropanol, chloroform, and 75% ethanol.

The detection and quantification of the p210 and p190 isoforms was performed using the Xgen RNA reference p210 and p190 kits, according to the manufacturer’s instructions (Mobius Life Science, Pinhais, Brazil). Both kits provide negative controls and standards (used for quantitative analysis). The analyses were performed in duplicate on a microplate, and for the final quantification of *BCR-ABL* p210 and p190, transcripts were normalized with control gene *ABL*.

The reactions were performed in an ABITM SDS 7300 Real-Time PCR (Thermo Fisher Scientific, Waltham, USA) under the following conditions: 50 cycles of 10 min at 50 °C, 5 min at 95 °C, 20 s at 95 °C, and 1 min at 60 °C. Data analysis followed the international standard with linear coefficient correction (*R^2^*): 98 ≤ *R^2^* ≤ 1000 and a slope from 3.0 to 3.4.

## 3. Results

A total of 1132 samples from CML patients were analyzed for diagnosis confirmation, treatment monitoring, and assessment of the molecular response. The qRT-PCR identified in two patients (0.17%) an increase in the *BCR-ABL*/*ABL* % of the p210 and p190 ratio (Table 1), which was indicative of a progression to BP. The time of progression occurred between 6 and 8 months after the CML diagnosis, and the two patients showed clinical worsening and were admitted to the Ophir Loyola Hospital for clinical monitoring.

The significant increase in *BCR-ABL* under imatinib and laboratory tests revealed leukocytosis, which ranged from 147,720.00/mm^3^ (patient 1) to 328,720.00/mm^3^ (patient 2) and 55% to 85% of blasts (Table 2), reflecting the progression to blast crisis. Overall survival (OS) averaged 9.5 months and disease-free survival (DFS) averaged 4.5 (Table 2). The DFS, which refers to the period of response to relapse, in this study, coincided with the rapid increase in blasts.

Both patients showed, along with the co-expression of the *BCR-ABL* transcript isoforms p210 and p190, the concomitant presence of a double Ph+ chromosome which was observed by conventional cytogenetics and confirmed by fluorescent in situ hybridization. The conventional cytogenetic study showed that patient 1 had a monosomy 17 as an additional chromosomal alteration. FISH analysis was used to confirm the presence of double *BCR-ABL* (Figure 1A,B).

The microscopic findings in both patients allowed us to visualize a hypocellular granulocytic and erythrocytic series, a normocellular lymphoplasmacytic series, and a megakaryocytic series, which are not reproduced here. The myelogram revealed the presence of immature cells from the granulocytic, erythroid, and lymphoplasmacytic lineages. Biochemical analyses showed an increase in lactic dehydrogenase and protein C, indicating cell damage and the presence of infections/inflammations, respectively. The elevation of lactic dehydrogenase is common in leukemias and lymphomas, and its activity increased when the disease worsened.

## 4. Discussion

The Ph+ chromosome is present in more than 90% of CML patients and, in the transition from the chronic to the blast phase, additional chromosomal abnormalities to the Ph+ chromosome are revealed via cytogenetic and molecular analysis [22], such as with the two patients who progressed to blast crisis, who presented in this study a karyotype with a double Ph+ chromosome.

One of the two patients with a double Ph+ chromosome had an additional third chromosomal abnormality represented by the monosomy of chromosome 17 (patient 1). Due to the impact on survival, all two alterations (Ph+, −17) present a high risk of death [23]. Additionally, Ph+ doubling is associated with an overexpression of the BCR-ABL protein and is considered to be a complex karyotype. Therefore, it is indicative of a poor prognosis in CML [24,25], as observed in this study, where the OS of the two patients ranged from 7 months to 12 months. Chromosome 17 monosomy implies the loss of the *TP53* gene. The deletion of the *TP53* gene is more frequent in solid tumors and occurs in 20% of patients in the blast phase of CML and is interpreted as a genetic marker of poor prognosis because monosomy may be part of the mechanism of loss of heterozygosity of this suppressor gene tumor [26,27].

The emergence of additional chromosomal abnormalities implies the development of genome instability and may be a reason for resistance to tyrosine kinase inhibitors (TKIs) [22,28,29,30,31,32]. For this reason, hematopoietic stem cell transplantation is indicated for patients with CML who have developed a burst phase and blast crisis and who do not respond to TKI therapy [33,34,35]. The patients in this study were resistant to TKIS, and there was no time for a transplant because they died.

Of the 1132 patients with CML analyzed by qRT-PCR for the transcript *BCR-ABL*, two patients (0.17%) who progressed to blast crisis presented a co-expression of the isoforms p210 and p190 (Table 1), and this percentage is lower than the 0.66% corresponding to three patients who progressed to BP in an Italian study from a cohort of 450 patients with CML, of which 29 patients had concomitant transcripts p190 and p210 [36], and it is also lower than the 7.05% of CML cases (six patients) that expressed the two main *BCR-ABL* isoforms in northeastern Iran, but this study is not a good point of reference because the co-expression found was not related to the development of blast crisis [37].

It should be noted that, to our knowledge, this is the first report in the literature of patients who co-expressed the two isoforms of the *BCR-ABL* transcript and concomitantly presented Ph+ chromosome duplication (Table 1). This combination of alterations is likely to trigger a more pronounced clonal evolution and exacerbation of CML.

The double Ph+ involves, therefore, the overexpression of the BCR-ABL protein, is described as a complex phenotype, has poor prognosis, is resistant to chemotherapy with TKIs, presents genomic transformations and instabilities, and occurs in the explosion phase [14], which agrees with what we observed in this study.

The vast majority of CML patients (>85%) are diagnosed in the chronic phase, and within 5 to 7 years there are cases that evolve to blast crisis [38]. In this study, the evolution to blast crisis was significantly earlier, since DFS was on average 4.5 months, and this phenomenon may be related to the presence of the double Ph+ chromosome and the fact that the BCR-ABL/ABL ratio % for p210 and p190 increased from diagnosis to blast crisis (Table 1), which means that there was no molecular improvement.

Studies have evaluated the impact of the p190 isoform on clinical features and disease outcomes in patients with CML [32,39,40,41]. Patients expressing other isoforms of BCR-ABL proteins, other than p210, failed to obtain a therapeutic response when submitted to imatinib mesylate [42].

In a case description by Junmei et al. [32], a 25-year-old patient with imatinib mesylate resistance was treated with dasatinib and later developed blast crisis. Both *BCR-ABL* fusion transcripts, p210 and p190, were positive, but after two months, the patient presented hematological remission. This is unlike the two patients in this study, whose overall survival was 7 and 12 months (Table 2).

Agirre et al., 2005 [43], reported a case of a patient with CML, positive for the Ph+ chromosome, with the presence of *BCR-ABL1* transcripts, p190 and p210. After treatment with imatinib mesylate, the p210 transcript was not detected, whereas p190 was still present 6 months after the initiation of therapy and in the progression to the blast phase. The absence of post-treatment p210 transcript indicates the existence of two clones, the p210 that responded to imatinib and the p190 clone that showed resistance to this drug. In the two patients in our study, there was probably only one clone, since the two transcripts, p190 and p210, had their expression increased from the beginning of therapy and in the evolution to the blast crisis.

The blast crisis is one of the remaining challenges in the CML. The concomitant analysis of *BCR-ABL* transcripts and chromosomal abnormalities in addition to the Ph+ chromosome could allow for an early recognition of impending blast proliferation and a timely change of treatment. Cytogenetic monitoring is important when disease progression or response to therapy is poor [23].

## 5. Conclusions

The presence of the p190 transcript and the double Ph+ chromosome in CML may be related to the vertiginous progression of the disease.

## Figures and Tables

**Figure 1 genes-13-00580-f001:**
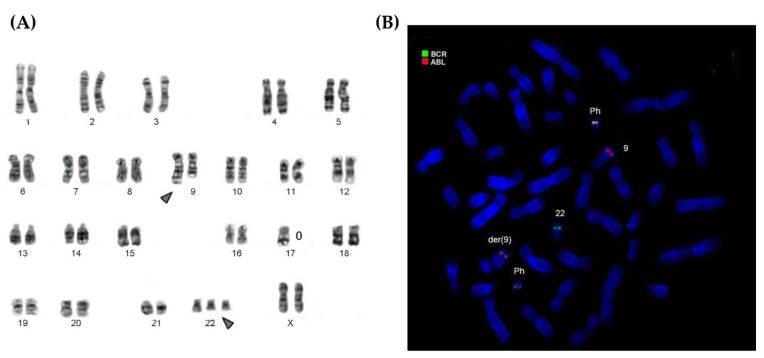
Cytogenetic analysis of human metaphase cells with double Ph+ chromosome magnified 1000 times (patient 1). (**A**) GTG banding karyotype, arrows point to (9;22), +der (22)t(9;22), and 0 represents monosomy of chromosome 17. (**B**) Metaphase submitted to FISH: chromosome 9, the 22, the two Ph+, and the derivative chromosome 9 (der (9)) are labelled in the micrograph.

**Table 1 genes-13-00580-t001:** Molecular and cytogenetic analysis of patients.

Patient	Diagnosis		Blast Crisis		
	Isoforms *BCR-ABL/ABL* %		Isoforms *BCR-ABL/ABL* %		Karyotype
	p210	p190	p210	p190	20 Metaphases *
1	108.54	73.68	125.16	84.74	46, XX, t(9;22), −17, +der(22)t(9;22)
2	100.59	70.35	120.46	80.86	47, XY, t(9;22), +der(22)t(9;22)

Legend: * number of metaphases.

**Table 2 genes-13-00580-t002:** Patient’s clinical data profile.

Patient	Age	Leukocytes	Blasts	OS *	DFS **
	Years	(mm^3^)	(%)	Months	Months
1	44	328,720.00	85	7	4
2	45	147,720.00	55	12	5

Legend: * overall Survival; ** disease-free survival.

## Data Availability

Not applicable.

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
