# Peer review of "Chronic Myelogenous Leukemia with Double Philadelphia Chromosome and Coexpression of p210 and p190 Fusion Transcripts"

_genes, 2022, doi:10.3390/genes13040580_

Round 1

Reviewer 1 Report

The article deals with CML and it is clearly written. However, I identified several issues.

1) In Introduction I lack information about double Philadelphia chromosome. How often it occurs? Why it is significant?

2) In section Patients and Methods you claim, that you have 5 patients. However, in result section you started with "Of the 1132 patients ...". It is not clear how many patients were used for individual method a why.

3) In conclusion you stated that monitoring imatinib mesylate resistance can be significant  to guide clinical management and predict prognosis. I would like to know how many patients were treated with imatinib mesylate in your patients' group and if only the five exhibit "resistance".

4) Why only 51 patients had bone marrow analyzed by cytogenetics? That was because their evolved to BP?

5) On lines 196-202 you claim that Ph+ doubling is a factor, that predicts resistance to imatinib mesylate. However, I have not found any information about this type of resistance in your references (24,25).

6)Reference 36 deals with 29 patients that co-expressed p210 and p190. If we think only patients with blast crisis there were 3 patients not one as you claim.

Author Response

RESPONSE TO REVIEWER

Dear reviewer, my co-authors and I thank you for the suggestions made during this high-quality review and then we present the answers to the questions.

We inform that with the reviews and suggestions, we were able to improve the idea presented by our work and we appreciate the opportunity. We hope this review has left the article suitable for publication in this high-impact and quality journal.

REVISIONS REQUESTED

The article deals with CML and it is clearly written. However, I identified several issues.

1. In Introduction I lack information about double Philadelphia chromosome. How often it occurs? Why it is significant?

R = This information was included in the introduction

2. In section Patients and Methods you claim, that you have 5 patients. However, in result section you started with "Of the 1132 patients ...". It is not clear how many patients were used for individual method a why.

R = This information was clarified in the article, because at the request of reviewer #2, the study was transformed into a case report of two patients with both isoforms (p190 and p210) and a duplicated Ph chromosome.

3. In conclusion you stated that monitoring imatinib mesylate resistance can be significant  to guide clinical management and predict prognosis. I would like to know how many patients were treated with imatinib mesylate in your patients' group and if only the five exhibit "resistance".

R= The question is appropriate. After the manuscript was transformed into a case report, we approached only two patients, both treated with imatinib mesylate and both showed resistance to this drug. This was clarified in the results and discussion of the manuscript.

4. Why only 51 patients had bone marrow analyzed by cytogenetics? That was because their evolved to BP?

R = In the affirmative, only 51 patients were analyzed by conventional cytogenetics due to the evolution of the clinical picture to BP. However, when transforming the study into a case report, the writing of the manuscript focused exclusively on the two patients who presented both p210 and p190 isoforms with the concomitant presence of double Philadelphia.

5. On lines 196-202 you claim that Ph+ doubling is a factor, that predicts resistance to imatinib mesylate. However, I have not found any information about this type of resistance in your references (24,25).

R = The referee is right and we apologize, this information was removed from the paragraph of references 24 and 25. This change does not affect the construction of the manuscript because this information was inserted in other paragraphs and referenced with correct bibliography.

6. Reference 36 deals with 29 patients that co-expressed p210 and p190. If we think only patients with blast crisis there were 3 patients not one as you claim.

R = We appreciate the referee's observation, it is completely correct. The paragraph has been modified to insert the true information.

Reviewer 2 Report

This is a study on five patients in blastic phase CML with Ph+ chromosome duplication +/- co-existence of P210 and P190 isoforms at diagnosis (before progression to blastic phase). Patients were identified in either of two ways: cytogenetic analysis of patients in blastic phase or BCR-ABL typing in a large cohort of newly diagnosed CML patients. While the occurrence of both Ph duplication and P210/P190 coexistence is worth reporting, the study has some flaws. First, it lacks originality: Association of high risk additional cytogenetic abnormalities, like Ph+ duplication, with TKI resistance and blastic phase is known. This is also true for the far less frequent P210/P190 coexistence. Regarding the latter, the reference PMID: 15949566 is lacking. Second, the clinical session is inconsistent with current therapy as there is no indication of monotherapy with TKIs of any generation or any dose in BP CML and patients did not receive induction chemotherapy, therefore conveying an erroneous clinical message. Imatinib resistance in BP is well documented, therefore it is not surprising that the BCR-ABL/ABL ratio increased with time. Reporting on response at intermediate time-points, before blastic phase, would be interesting. Third, the study does not offer documentation on clonal evolution.

In my opinion, the study should restrict to biological findings and focus, as a case report, on the two patients with both isoforms and a duplicated Ph chromosome.

Author Response

Dear reviewer, my co-authors and I thank you for the suggestions made during this high-quality review and then we present the answers to the questions.

We inform that with the reviews and suggestions, we were able to improve the idea presented by our work and we appreciate the opportunity. We hope this review has left the article suitable for publication in this high-impact and quality journal.

REVISIONS REQUESTED

This is a study on five patients in blastic phase CML with Ph+ chromosome duplication +/- co-existence of P210 and P190 isoforms at diagnosis (before progression to blastic phase). Patients were identified in either of two ways: cytogenetic analysis of patients in blastic phase or BCR-ABL typing in a large cohort of newly diagnosed CML patients. While the occurrence of both Ph duplication and P210/P190 coexistence is worth reporting, the study has some flaws. First, it lacks originality: Association of high risk additional cytogenetic abnormalities, like Ph+ duplication, with TKI resistance and blastic phase is known. This is also true for the far less frequent P210/P190 coexistence. Regarding the latter, the reference PMID: 15949566 is lacking. Second, the clinical session is inconsistent with current therapy as there is no indication of monotherapy with TKIs of any generation or any dose in BP CML and patients did not receive induction chemotherapy, therefore conveying an erroneous clinical message. Imatinib resistance in BP is well documented, therefore it is not surprising that the BCR-ABL/ABL ratio increased with time. Reporting on response at intermediate time-points, before blastic phase, would be interesting. Third, the study does not offer documentation on clonal evolution.

In my opinion, the study should restrict to biological findings and focus, as a case report, on the two patients with both isoforms and a duplicated Ph chromosome.

R = We followed the recommendation of the referee, whose observations are adequate, transferred the article to a case report format and narrowed the focus of the manuscript to biological findings.

Thanks for the suggestion of the PMID reference: 15949566, it fit perfectly into the discussion.

Reviewer 3 Report

da Cruz et al present a case series reporting CML patients with double Ph+ chromosome including two individuals that express both the p210 and p190 BCR-ABL1 isoform. This has not previously been reported in the literature.

This finding does have some novelty although the clinical significance of it is unclear. The major concern with this manuscript is that it is very difficult to understand and has multiple typos. This makes it quite challenging to interpret the results or the conclusions, and decreases enthusiasm for publication in its current form. I would recommend substantial professional editing prior to consideration.

Minor concerns:

Multiple typos including the spelling of “with” in the title and “chromosome. for BP, express..”

“the 5' portion. from the sequences”

The timing of the molecular studies relative to progression to blast crisis is not clear.

“the patients developed severe pain in the gingiva and fever, which changed the medical approach to substitution of imatinib mesylate  for Dasatinib (100 mg/day)”

Was this for all patients or just one? Not clear.

“Thus, monitoring imatinib mesylate resistance in CML patients, especially for blast crisis CML, can be significant to guide clinical management and predict prognosis.”

This is true, but is not a conclusion which is particularly supported by the findings presented here.

TKI resistance is frequently accompanied by ABL kinase domain mutations. Was this performed?

Author Response

Dear reviewer, my co-authors and I thank you for the suggestions made during this high-quality review and then we present the answers to the questions.

We inform that with the reviews and suggestions, we were able to improve the idea presented by our work and we appreciate the opportunity. We hope this review has left the article suitable for publication in this high-impact and quality journal.

REVISIONS REQUESTED

da Cruz et al present a case series reporting CML patients with double Ph+ chromosome including two individuals that express both the p210 and p190 BCR-ABL1 isoform. This has not previously been reported in the literature.

This finding does have some novelty although the clinical significance of it is unclear. The major concern with this manuscript is that it is very difficult to understand and has multiple typos. This makes it quite challenging to interpret the results or the conclusions, and decreases enthusiasm for publication in its current form. I would recommend substantial professional editing prior to consideration.

R = The writing of the manuscript was revised and edited.
Regarding the clinical significance, the manuscript was modified at the request of reviewer #2, who requested that the study should be restricted to biological findings and focus, as a case report, on the two patients with both isoforms and the duplicated Ph chromosome. 

Minor concerns:

Multiple typos including the spelling of “with” in the title and “chromosome. for BP, express..”

“the 5' portion. from the sequences”

R = We apologize for the errors, all have been corrected. 

The timing of the molecular studies relative to progression to blast crisis is not clear.

R =  Added in the first paragraph of the results item

“the patients developed severe pain in the gingiva and fever, which changed the medical approach to substitution of imatinib mesylate  for Dasatinib (100 mg/day)”

Was this for all patients or just one? Not clear.

R = This information was taken from the article since, as previously mentioned, at the request of reviewer #2, the study was restricted to biological findings and was transformed into a case report of two patients with both isoforms and a duplicated Ph chromosome. 

“Thus, monitoring imatinib mesylate resistance in CML patients, especially for blast crisis CML, can be significant to guide clinical management and predict prognosis.”

This is true, but is not a conclusion which is particularly supported by the findings presented here.

R = The referee is right; this conclusion was withdrawn.

TKI resistance is frequently accompanied by ABL kinase domain mutations. Was this performed?

R = The reviewer question is appropriate; we did not study mutations in the Kinase domain of the ABL gene and for this reason we modified this paragraph in the manuscript.

Round 2

Reviewer 1 Report

The manuscript changed from article to case report. It is now clear and methods are adequately described. The conclusion is supported by results. I have only one comment.

On lines 139-140 please change patient 3 and 4 for patient 1 and 2. It is a remnant from original version that should be corrected.

Author Response

RESPONSE TO REVIEWER

Dear reviewer, my co-authors and I thank you for the suggestions made during this high-quality review and then we present the answers to the questions.

We inform that with the reviews and suggestions, we were able to improve the idea presented by our work and we appreciate the opportunity. We hope this review has left the article suitable for publication in this high-impact and quality journal.

REVIEWER 1

The manuscript changed from article to case report. It is now clear and methods are adequately described. The conclusion is supported by results. I have only one comment.

On lines 139-140 please change patient 3 and 4 for patient 1 and 2. It is a remnant from original version that should be corrected.

R = We appreciate the referee's observation. The paragraph has been modified to insert the true information.

Reviewer 2 Report

The authors revised the manuscript by adopting a case report format focusing on two patients with isoform coexpression and Ph+ duplication.

Major comments:

No clonal evolution in documented, the term should therefore not be used in the title.

There are many repetitions, for example on the size of the initial sample (1132 patients) and the identification of two patients with the above characteristics. 

There are inconsistencies which are remnants from the previous format, for example mentions to patient 3 and patient 4 who were no longer included in the current format.

In conclusion, the case report could be considered for publication only after revision of the above points.

Author Response

RESPONSE TO REVIEWER

Dear reviewer, my co-authors and I thank you for the suggestions made during this high-quality review and then we present the answers to the questions.

We inform that with the reviews and suggestions, we were able to improve the idea presented by our work and we appreciate the opportunity. We hope this review has left the article suitable for publication in this high-impact and quality journal.

REVIEWER 2

Major comments:

No clonal evolution in documented, the term should therefore not be used in the title.

R = Dear reviwer, the suggestion was observed, and we created a new title:  

“Chronic Myelogenous Leukemia With Double Philadelphia Chromosome and Coexpression of p210 and p190 Fusion Transcripts”

There are many repetitions, for example on the size of the initial sample (1132 patients) and the identification of two patients with the above characteristics. 

R = The corrections were made.

There are inconsistencies which are remnants from the previous format, for example mentions to patient 3 and patient 4 who were no longer included in the current format.

R = The corrections were made.

Round 3

Reviewer 1 Report

The manuscript is now OK

Author Response

Dear reviewer, my co-authors and I thank you for the suggestions made during this high-quality review and then we present the answers to the questions.

We inform that with the reviews and suggestions, we were able to improve the idea presented by our work and we appreciate the opportunity. We hope this review has left the article suitable for publication in this high-impact and quality journal.

Reviewer 2 Report

Some corrections must still be made in the esults:

Two patients out of xx patients (0,17%)... Actually, the sentence of lines 152-155 should be used to introduce the results.

The significant increase of BCR-ABL under imatinib

The microscopic findings in the five patients

Better reformulate the microscopic findings

Author Response

(The authors gave the same response as above.)
